# Design of Quad-Band Bandpass Filter Using Dual-Mode SLRs and Coupled-Line for DCS/WLAN/WiMAX and 5G Applications

**DOI:** 10.3390/mi13050700

**Published:** 2022-04-29

**Authors:** Sugchai Tantiviwat, Siti Zuraidah Ibrahim, Mohammad Shahrazel Razalli, Ping Jack Soh

**Affiliations:** 1Faculty of Industrial Education and Technology, Rajamangala University of Technology Srivijaya, Muang District, Songkhla 90000, Thailand; 2Faculty of Electronic Engineering Technology, Universiti Malaysia Perlis (UniMAP), Arau 02600, Malaysia; sitizuraidah@unimap.edu.my (S.Z.I.); shahrazel@unimap.edu.my (M.S.R.); pjsoh@unimap.edu.my (P.J.S.)

**Keywords:** multi-band BPFs, stub-loaded resonator (SLR), dual-mode resonator

## Abstract

A design of a microstrip quad-band BPF with flexibly controlled bandwidth is presented in this paper. Two dual-mode short-circuited SLRs with a common via-hole are proposed, which are utilized to obtain the first and second passband, while the third passband is generated by implementing the second-order half-wavelength coupled-line resonator. Another dual-mode open-circuited SLR can be operated at the fourth passband. The proposed quad-band BPF is centered at 1.80/2.45/3.50/4.90 GHz for DCS/WLAN/WiMAX and 5G applications. By appropriately choosing the lengths of the four sets of resonators, all passbands can be fully varied independently with minimal effect on other passbands. Moreover, the bandwidth of each passband can be flexibly controlled by tuning the coupling parameters. The dimension of the fabricated proposed filter is about 0.12 × 0.20 λg, indicating the compactness of the design, whereas the measurements are in good agreement with the simulated results. The measured S11 are at least 12 dB in the four passbands. The passbands S21 are approximately 0.65, 1.42, 0.78, and 1.20 dB, which exhibit low insertion loss at the passband frequency of the first, second, third, and fourth passband, respectively.

## 1. Introduction

Multi-band bandpass filters (BPFs) have been developing rapidly in line with the importance of such filters for multi-service wireless communication systems in recent years [1,2,3]. To provide for the needs of multiple-passband applications such as in DCS/WLAN/WiMAX/5G, quad-band BPFs have been commonly proposed and studied [4,5,6,7,8,9,10,11]. Commonly, the procedure for designing multi-band BPFs can be implemented with parallel and/or cross coupling using stepped impedance resonators (SIRs) [4,5]. By using such techniques, the multi-band feature can be achieved by adjusting the impedance and length ratios. However, the resulting resonant frequencies are not independent and are mainly caused by the characteristics of SIRs which depend on the frequency ratios.

Moreover, most conventional multiple-band BPFs with single-mode resonators are large in size. To alleviate this aspect, multiple-mode stub-loaded resonators (SLRs) have been explored, due to their ability to control resonant frequencies [6,7,8,9,10,11]. Such multiple-band BPFs are coupled with cascade and/or parallel resonators. For instance, the set of resonators with open- and short-circuit stub-loaded resonators applied in the quad-band filter design in [6,7,8]. However, the final structure is relatively large due to the need for several resonators, which demands more short-circuit via-holes. Furthermore, these methods are not flexible in controlling the bandwidth of each passband. In [9,10,11], the multiple-mode technique is used to design the quad-band BPFs. Although this structure is compact, multi-band BPFs can be designed using this resonator to generate multi-mode resonant frequencies. However, some resonant frequencies within each passband are still unable to be independently controlled. In addition, folded asymmetric stub-loaded resonators were reported, which are applied to balanced quad-band BPFs [12]. For quad-band operation, unfortunately, there is little research work reported on quad-band BPFs.

In this paper, a quad-band BPF which is compact in size, with flexibly controlled bandwidths and independently controlled passbands using simple resonator structures, is presented. To generate two passbands with independently controllable bandwidths, two short-circuit dual-mode resonators operating at different frequencies are utilized. The proposed BPF has compact circuit size and strong design feasibility since the arrangement of the coupled dual-mode SIRs and the quad-bands can be easily deter-mined by properly tuning the dimension of the independent resonators. To achieve compactness, both short-circuit loaded stubs (SLRs) are designed to share a common via-hole. Half-wavelength resonators are designed at the input/output ports and are located beside two single-mode resonators and the open-circuit dual-mode SLR. Such a configuration successfully generated the third and fourth passband responses, respectively.

## 2. Analysis of the Second-Order Quad-Band BPF

Figure 1 illustrates the layout of the proposed second-order quad-band BPF which is designed on a Diclad Arlon 880 substrate with a thickness of 0.80 mm, relative dielectric constant of 2.20, and loss tangent of 0.009. The proposed filter consists of two dual-mode short-circuit stub-loaded resonators that share a common via-hole, a dual-mode open-circuit stub-loaded resonator, and a two half-wavelength coupled-line resonators, respectively.

Figure 2a shows the layout of the dual-mode stub-loaded resonator used to enable operation of this filter in the first and second passbands. It consists of a half-wavelength open ended stub and a short-circuit stub. Parameters *Z*_1_, *Z*_2_ are the characteristic impedances, and *L*_1_, *L*_2_ denote the lengths of the microstrip line and short-circuit stub, respectively. Since the proposed structure is symmetrical, the odd/even-mode analysis method can be used to characterize the resonant frequencies. Its odd- and even-mode equivalent circuits are shown in Figure 2b,c, respectively. Under the condition of *Z*_1_ = 2*Z*_2_, the resonant frequencies of the dual-mode resonator can be approximated by
*f*_odd1_, *f*_odd2_, = *cθ*_1_/(2π*L*_1_ × *ε_eff_*^1/2^) = *c*/(2*L*_1_ × *ε_eff_*^1/2^)(1)
*f*_even1_, *f*_even2_, = *c*/(2 × (*L*_1_ + *L*_2_) × *ε_eff_*^1/2^)(2)
where *θ* = *βL*_1_ is the electrical length of the open stub, *c* is the speed of light in free space, and *ε_eff_* denotes the effective dielectric constant of the substrate.

Figure 3a shows the layout of the dual-mode open-circuit stub-loaded resonator de-signed for operation of this filter in the fourth passband. The odd-mode and even-mode equivalent circuits are shown in Figure 3b,c, respectively. Under the condition of *Z*_4_ = 2*Z*_5_ = *Z*_6_, the resonant frequencies of the dual-mode resonator can be approximated by
*f*_odd4_ *= cθ*_4_/(2π*L*_4_ × *ε_eff_*^1/2^) = *c*/(2*L*_4_ × *ε_eff_*^1/2^)(3)
*f*_even4_ = *c*/((*L*_4_ + 2(*L*_5_ + *L*_6_)) × *ε_eff_*^1/2^)(4)

The combinations of Equations (1) to (4), resonant frequencies of the first, second, and fourth passband can be obtained with *f*_even1_ < *f*_odd1_ < *f*_even2_ < *f*_odd2_ < *f*_even4_ < *f*_odd4_. Thus, the even-mode resonant frequencies of the proposed resonator, which is designed to operate in the first and second passband, can be controlled by adjusting the short-stub length, *L*_2_*^I^* and *L*_2_*^II^*. The resonant characteristics of the two even-mode resonant frequencies are analyzed using parametric studies shown in Figure 4a,b, with the superscripts *I* and *II* denoting the first and second passband, respectively. Alternatively, influencing the length of *L*_1_ on the odd-mode resonant frequency, *f*_odd_, and the even-mode resonant frequency, *f*_even_, which is used as the first and second passband frequencies in this filter, can also be found by analyzing Equations (1) and (2). Figure 4c,d illustrates the effects of *L*_1_ on the passband frequencies. As the lengths of transmission lines *L*_1_*^I^* and *L*_1_*^II^* increase, the first and second passband is lowered, whereas changes in the other passbands remain minimal.

On the other hand, the operation of the open-circuit SLR designed for the fourth passband can be controlled by modifying the open-circuit stub length (*L*_5_ + *L*_6_). The resonant characteristics of the even-mode resonant frequency (*f*_even4_) and odd-mode resonant frequency (*f*_odd4_) are analyzed by varying the values of *L*_4_ and *L*_6_, and maintaining a fixed *L*_5_ value, as shown in Figure 4e. Meanwhile, the third passband can be controlled by varying the length of *L*_3_, since the half-wavelength resonator determines the third passband. The effects of the variation in the length of *L*_3_ with a fixed value of the gap between resonators (*g*_1_) can be observed in the simulated insertion loss (*S*_21_), and is shown in Figure 4f, respectively.

Figure 4a,b shows the preliminary simulated insertion loss *S*_21_ with different lengths of *L*_2_*^I^* and *L*_2_*^II^*, when the dimension of the via-hole is fixed at 0.60 mm. Clearly, the main resonant frequencies of first passband and second passband can be controlled by varying the lengths of *L*_1_*^I^* and *L*_1_*^II^*, respectively. As observed in Figure 4a–d, the even-mode can be shifted when the odd-mode of the first and second passband remained unchanged. Moreover, both the even- and odd-mode resonant frequencies of the fourth passband can be separately adjusted by varying the lengths of *L*_4_, *L*_5_, and *L*_6_. Finally, the operation in the third passband is produced by the half-wavelength resonator based on the length of *L*_3_, as illustrated in Figure 4f. Most importantly, all the passbands of the proposed filter are able to be flexibly controlled without affecting operation at all other passband frequencies.

## 3. Quad-Band BPF Design

The proposed quad-band BPF is implemented based on the analysis in the previous section. The center frequencies of the four passbands of the proposed BPF were located at 1.80, 2.45, 3.50, and 4.90 GHz. The fractional bandwidths (FBW) were determined as 7.29%, 4.08%, 5.71%, and 4.08%, respectively. To determine the fractional bandwidth, coupling coefficient (*M*_ij_) and external quality factor (*Q*_e_) were calculated based on the method presented in [12], where i, j are resonant frequencies of second-order passband frequency. The calculated values are summarized in Table 1. The quad-passband BPF shown in Figure 1 was designed using a well-known classical filter analysis method in [13], where a second-order Chebyshev frequency response with a 0.1 dB ripple level was considered. The lumped circuit element values of the normalized low-pass prototype filter were found to be *g*_0_ = 1, *g*_1_ = 0.8431, *g*_2_ = 0.6220, and *g*_3_ = 1.3554, respectively.

The design procedure for the proposed filter is described as follows. Firstly, the resonators of the four passbands were designed, and their dimensions were optimized as follows: For the first passband at 1.80 GHz, the length *L*_1_*^I^* = 58.20, *L*_2_*^I^* = 1.10, and width *W*_1_*^I^* = 1.00, *W*_2_*^I^* = 2.00. For the second passband at 2.45 GHz, the length *L*_1_*^II^* = 44.95, *L*_2_*^II^* = 0.95, and width *W*_1_*^II^* = 1.00, *W*_2_*^II^* = 2.00. For the third passband at 3.50 GHz, the half-wavelength resonator was designed to resonate with a dimension of *L*_3_ = 33.75, *W*_3_ = 0.40. For the fourth passband at 4.90 GHz, the length *L*_4_ = 24.20, *L*_5_ = 0.95, *L*_6_ = 7.45, and width *W*_4_ = 0.50, *W*_5_ = 1.00, *W*_6_ = 0.50 (All dimensions are in mm).

Secondly, an appropriate value of the specific coupling coefficient and the external quality factors were determined as shown in Figure 5 and Figure 6. The simulated result of *g*_1_ is established in Figure 5a. Through this figure, the optimized *g*_1_ is found to be 0.30 mm. In turn, the optimized tapped line position (*t*) is found to be 13.20 mm according to the extracted value of external quality factor (*Q*_e_) versus tapped line (*t*) and is shown in Figure 6a.

The third step is to design the open-circuit dual-mode SLR for the fourth passband. After the dimension of the resonator is produced, the desired values of coupling coefficient (*M*_ij_) between the odd-mode and even-mode can be calculated based on the length of *L*_4_, *L*_5_, and *L*_6_, where the varied length of *L*_5_ is shown in Figure 5b while maintaining the length of *L*_6_ = 7.45 mm. The optimized values of external quality factor (*Q*_e_) based on *g*_2_ is found to be 0.25 mm, as shown in Figure 6b, which make it easy for fabrication.

The final step in the circuit design is to individually adjust and combine the two dual-mode resonators in the BPF. Upon determining the fractional bandwidth, the de-sired values of the coupling coefficient *M*_ij_ = (*f*^2^_odd_ − *f*^2^_even_)/(*f*^2^_odd_ + *f*^2^_even_) between the odd-mode and even-mode can be calculated based on the lengths of *L*_2_*^I^* and *L*_2_*^II^*. The relationship between *M*_ij_ and the *L*_2_ length of the first passband and second passband are shown in Figure 5c,d. The values of *Q*_e_ = *f*_0_/∆*f*_3dB_ for the input and output of the first and second passbands is shown in Figure 6c,d. In this case, the length of the input/output port and *L*_3_ are fixed, while varying the gap between port and resonators *d*_1_ and *d*_2_ to match the specific external quality factors of the first and second passbands. The optimized values of *d*_1_ and *d*_2_ are found to be 0.30 and 0.50 mm, respectively.

## 4. Implementation and Results

Figure 7a illustrates the fabricated quad-band BPF. The size of the filter is 14.70 × 24.65 mm, which is approximately 0.12 and 0.20 λ_g_, where λ_g_ is the guided wave-length at the center frequency of the first passband. The simulated and measured results are shown in Figure 7b. The measured covering desired passband return losses are at least 12 dB in the four passbands but the second passband has ripples in the measurement because of the fabrication based on shifting positions of the common via-hole between first and second passbands. The passband insertion losses are approximately 0.65, 1.42, 0.78, and 1.20 dB at the center frequency of the first, second-, third, and fourth passband, respectively. The sharp passband skirt near each passband edge is generated by the source/load coupling effect of the filter. The transmission zeroes (TZs) are generated at 1.40, 3.27, 4.15, and 5.10 GHz, respectively. As can be seen, the overall results of the measured results are in agreement with the simulated predictions. Table 2 shows the comparison of the proposed quad-band BPF with other reported BPFs. It can be seen that the proposed BPF outperforms the others in terms of a pair of sizes and minimal IL. The proposed filter can be easily fabricated and was seen to feature advantages such as low-insertion loss, compact size, and the individual control of passbands and bandwidths, since two parallel dual-mode resonators can be combined with a coupled-line resonator as a port input, including that it be coupled to the simultaneous open-circuit SLR dual-mode resonator.

## 5. Conclusions

This paper has presented the design of a quad-band BPF with flexibly controllable bandwidth. The proposed quad-band BPF has been designed using half-wavelength coupled-line resonators, two sets of short-circuit dual-mode SLRs with common via-hole, and open-circuit dual-mode SLRs. Such a configuration optimized the required space for the filter, with an overall dimension about 0.12 × 0.20 λg, which indicates design compactness. The four second-order resonators can be independently designed, and their bandwidths can be flexibly tuned without affecting other passbands. Most importantly, the proposed filter featured good selectivity and therefore is suitable for multi-band and multi-service applications.

## Figures and Tables

**Figure 1 micromachines-13-00700-f001:**
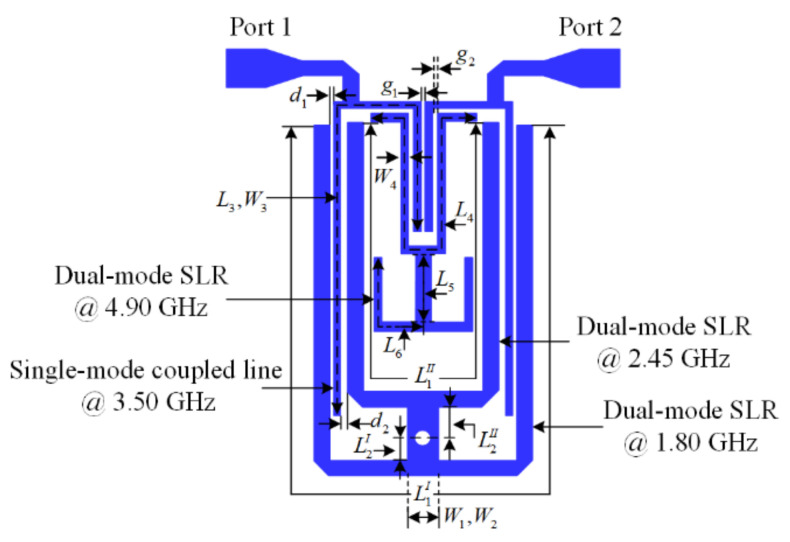
The layout of the proposed quad-band BPF.

**Figure 2 micromachines-13-00700-f002:**
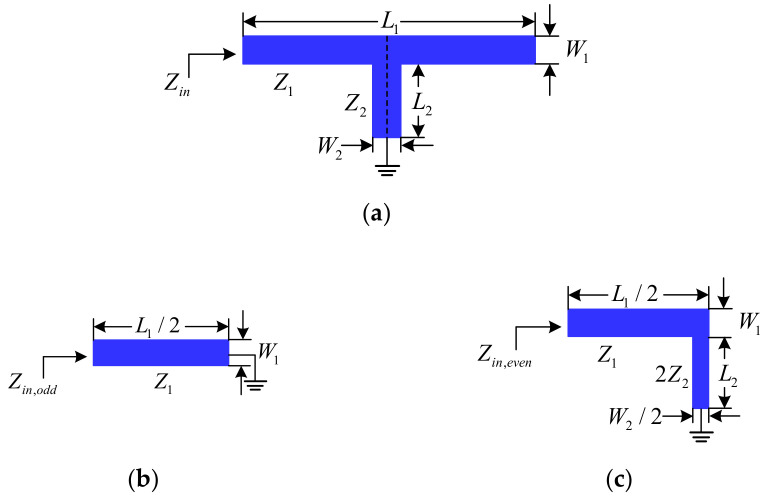
Schematic of the conventional dual-mode short-circuit stub-loaded resonator: (**a**) dual-mode resonator; (**b**) odd-mode; and (**c**) even-mode.

**Figure 3 micromachines-13-00700-f003:**
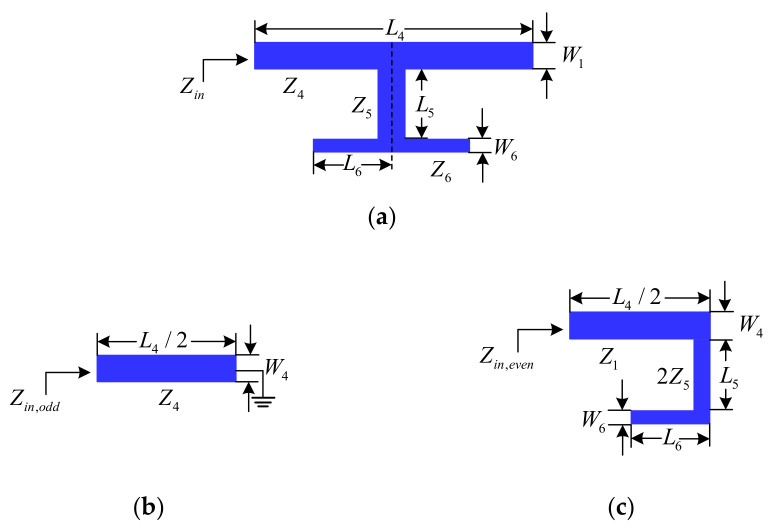
Schematic of the dual-mode open-circuit stub-loaded resonator: (**a**) dual-mode resonator; (**b**) odd-mode; and (**c**) even-mode.

**Figure 4 micromachines-13-00700-f004:**
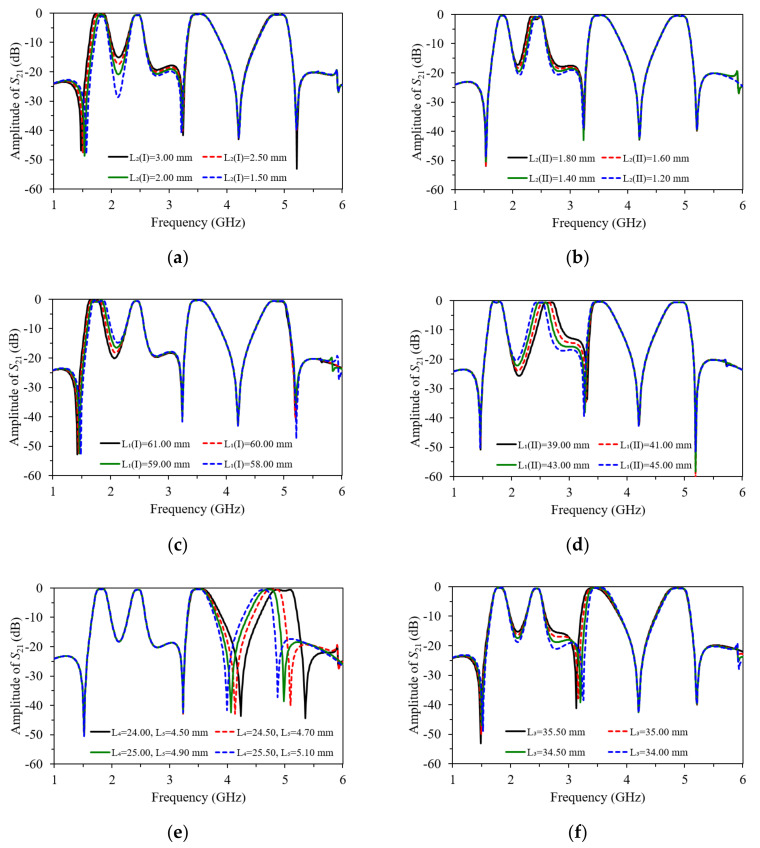
Variation on resonant frequency of the proposed resonator with different parameters: (**a**) *L*_2_*^I^*; (**b**) *L*_2_*^II^*; (**c**) *L*_1_*^I^*; (**d**) *L*_1_*^II^*; (**e**) *L*_4_ and *L*_5_; (**f**) *L*_3_.

**Figure 5 micromachines-13-00700-f005:**
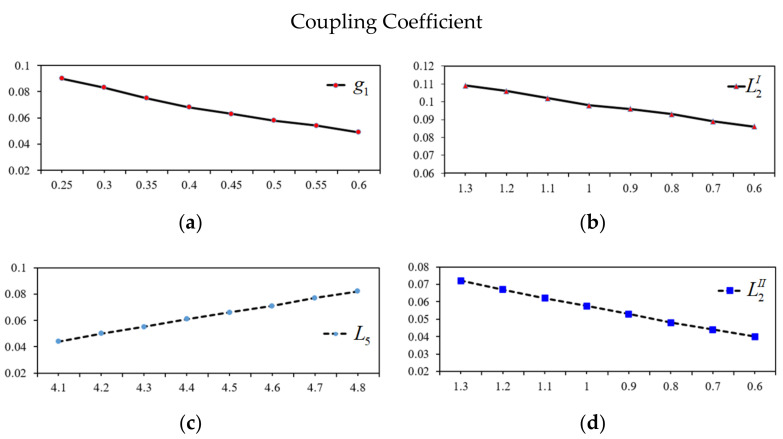
Coupling coefficient with variation of *g*_1_, *L*_5_ (*L*_6_ = 7.45 mm), *L*_2_*^I^*, and *L*_2_*^I^*^I^. (**a**) Distance gap *g*_1_ (mm). (**b**) The length of *L*_2_*^I^* (mm). (**c**) The length of *L*_5_ (mm). (**d**) The length of *L*_2_*^II^* (mm).

**Figure 6 micromachines-13-00700-f006:**
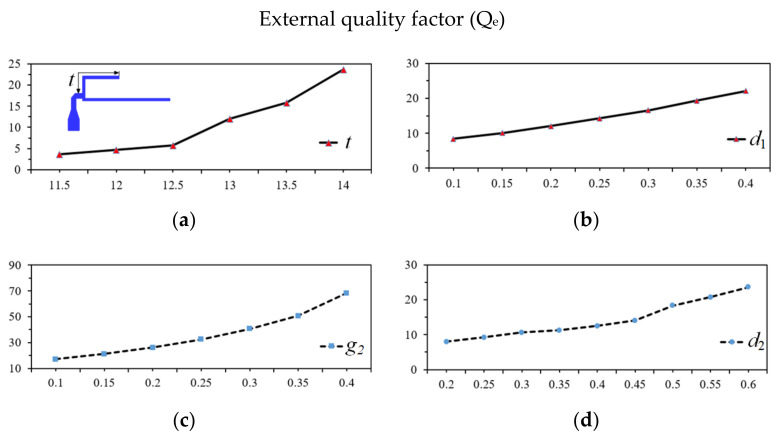
External quality factor with variation of *t*, *g*_2_, *d*_1_, and *d*_2_. (**a**) The tapped line position *t* (mm). (**b**) Coupling space *d*_1_ (mm). (**c**) Coupling space *g*_2_ (mm). (**d**) Coupling space *d*_2_ (mm).

**Figure 7 micromachines-13-00700-f007:**
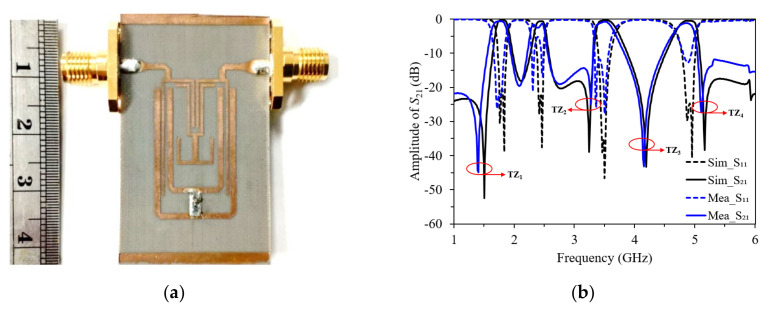
(**a**) The photograph of the proposed quad-band BPF; (**b**) simulated and measured results of the quad-band BPF.

**Table 1 micromachines-13-00700-t001:** The specifications of the proposed quad-band BPF.

Passband	Fractional BW (FBW)	Coupling Coefficient *(M*_ij_)	External Quality Factor *(Q*_e_)
First passband	7.29%	0.101	11.56
Second passband	4.08%	0.056	20.66
Third passband	5.71%	0.078	14.76
Fourth passband	4.08%	0.056	20.66

**Table 2 micromachines-13-00700-t002:** Comparison between the proposed quad-band BPF and other reported filters.

Ref.	*f*_0_ (GHz)	Measured 3 dB FBW (%)	IL (dB)	Substrate *h (*mm)/*ε*_r_	Size (λ_g_ × λ_g_)
[4]	2.4/3.5/5.2/6.8	6.4/9.4/3.8/4.9	0.5/1.3/1.3/1.0	0.787/2.20	0.20 × 0.20
[5]	2.4/3.3/5.38/6.48	3/6.41/3.7/4.56	1.9/1.6/3.5/3.2	0.508/3.50	0.28 × 0.18
[6]	1.5/2.5/3.6/4.6	5.5/12/11/4.3	1.98/1.74/3.58/3.4	0.508/3.55	0.30 × 0.30
[7]	1.8/2.4 /3.5/5.2	N/A	0.8/1.4 /1.7/2	0.787/2.20	0.22 × 0.10
[8]	2.4/3.5/5.2/5.8	6.7/7.2/6.9/5.3	2.0/1.9/1.9/1.96	0.508/3.66	0.16 × 0.25
[9]	1.8/2.45/3.5/5.5	6.7/4.2/3.7/14.8	1.5/1.7/2.3/1.8	0.81/3.38	0.19 × 0.15
[10]	1.46/2.6/4.2/5.25	8.6/4.5/5.2/4.2	1.5/1.9/1.7/2.0	0.81/3.38	0.14 × 0.12
[11]	0.91/1.47/2.46/3.54	7.90/10.50 /5.60/6.30	1.53/1.31/1.43/1.61	1.00/2.45	0.098 × 0.085
[12]	2.48/3.45/5.17/5.78	5.96/6.43/5.96/6.90	1.74/1.73/2.48/2.35	0.80/2.20	0.58 × 0.45
This work	1.80/2.45/3.50/4.90	13.74/4.52/8.33/5.80	0.65/1.42/0.78/1.20	0.80/2.20	0.12 × 0.20

(Note: *f*_0_ = center frequency; IL = insertion loss).

## Data Availability

Not applicable.

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
