# Peer review of "Design of Quad-Band Bandpass Filter Using Dual-Mode SLRs and Coupled-Line for DCS/WLAN/WiMAX and 5G Applications"

_micromachines, 2022, doi:10.3390/mi13050700_

Round 1
Reviewer 1 Report
A microstrip quad-band bandpass filter using dual-mode SLRs 2 and coupled-line for DCS/WLAN/WiMAX and 5G Applications is designed and implemented. Some suggestions are given as follows.
[1]Through detailed comparison with the literature listed in Table 2, the novelty of the designed filter should be highlighted.
[2] As for Fig. 7b, a certain degree of mismatch between simulation and measurement can still be observed, so more in-depth analysis should be given.
[3] Some gramatical errors can be found.
Author Response
Thank you for the correction, we have changes accordingly.
Point 1: Through detailed comparison with the literature listed in Table 2, the novelty of the designed filter should be highlighted.
Response 1: The novelty of the designed filter can be highlighted in line 182-187 as:
The proposed filter can be easily fabricated and was seen to feature advantages such as low-insertion loss, compact size, and the individual control of passbands and bandwidths since two parallel dual-mode resonators can be combined with a coupled-line resonator as port input, including that it be coupled to the simultaneous open circuit SLR dual-mode resonator.
Point 2: As for Fig. 7b, a certain degree of mismatch between simulation and measurement can still be observed, so more in-depth analysis should be given.
Response 2: The reason of mismatch between simulation and measurement is revised in line as:
The measured covering desired passband return losses are at least 12 dB in the four passbands but the second-passband has ripples in the measurement cause of the fabrication based on shifting position common via-hole between first- and second-passband. Line 174-176.
Point 3: Some gramatical errors can be found.
Response 3: The sentenses are rechecked and rewritten.
Reviewer 2 Report
- The presented background is not sufficient. It needs more elaborate discussion in the introduction.
- Only 12 references are not sufficient for a journal paper. Most of the cited papers are more than 5 years old.
- The novelty of this paper is not clear.
- In table 2, the authors have compared their work with some previous works published in the literature, but the cited papers are not the most recent publications. So, this comparison is not so appropriate to claim the advantages of the proposed work over the existing works.
Author Response
Thank you for the correction, we have changes accordingly.
Point 1: The presented background is not sufficient. It needs more elaborate discussion in the introduction.
Response 1: The background are rewritten. Line 46-48, 52-55.
Point 2: Only 12 references are not sufficient for a journal paper. Most of the cited papers are more than 5 years old.
Response 2: Unfortunately, there is little research work reported on quad-band BPF. However, the references are updated and rewritten. Line 46-48.
Point 3: The novelty of this paper is not clear.
Response 3: The detail of the additional novelty in this paper is inserted in line 182-187.
Point 4: In table 2, the authors have compared their work with some previous works published in the literature, but the cited papers are not the most recent publications. So, this comparison is not so appropriate to claim the advantages of the proposed work over the existing works.
Response 4: The references are updated and rewritten in Table 2 for comparison.

Reviewer 3 Report
The reviewer appreciate the work and results achieved by the authors. At the same time, the reviewer has the following comments and concerns:
- The method to find the design dimensions must be explained with the details.
- A perfect odd and even mode analysis (mathematic method) must be added.
- Comparison with the previous works in Table 2 must include harmonics sharpness, FBW and return losses.
- It seems that the presented filter cannot attenuate the harmonics well. Add a method to suppress the harmonics
- There is not a good isolation between the first and second channels.
- Recent references must be added. The number of references is not enough.
- The parameters, e.g., f, S21, Z, c, etc., should be written with italic font, whereas the dimension should be kept with regular font.
- Center frequency should be abbreviated as fc, not CF.
Author Response
Thank you so much for the correction, we have changes accordingly.
Point 1: The method to find the design dimensions must be explained with the details.
Response 1: The design dimensions are explained based on reference [13] in line 158-161.
Point 2: A perfect odd and even mode analysis (mathematic method) must be added..
Response 2: The design dimensions are added based on reference [13] in line 158-161.
Point 3: Comparison with the previous works in Table 2 must include harmonics sharpness, FBW and return losses.
Response 3: The FBWs are added for comparison in table 2. The return losses of some previous works in reference are not reported with whole passbands.
Point 4: It seems that the presented filter cannot attenuate the harmonics well. Add a method to suppress the harmonics
Response 4: The technique for attenuate the harmonics, the multi-order spurious mode suppression can be improved but this method increase the complex variables.
Point 5: There is not a good isolation between the first and second channels.
Response 5: The isolation between the first and second channels are desinged. The results is not good because the transmission zero near the first-, and second-passband.
Point 6: Recent references must be added. The number of references is not enough.
Response 6: Unfortunately, there is little research work reported on quad-band BPF. However, the references are updated and rewritten (Line 46-48), and the reference is added for comparison in table 2
Point 7: The parameters, e.g., f, S21, Z, c, etc., should be written with italic font, whereas the dimension should be kept with regular font.
Response 7: The details are rewritten. Thank you for the correction, we have changes accordingly.
Point 8: Center frequency should be abbreviated as fc, not CF.
Response 8: The details are rewritten to fo in table 2.

Round 2
Reviewer 1 Report
I think that the manuscript can be published in present form.
Reviewer 3 Report
The authors have addressed all my comments for this paper and answered the technical questions I have for this method. The paper has been improved after revision. Though the proposed Quad-band Bandpass Filter suffers from (1) low isolation between the first and second channels, and (2) small harmonics attenuation, the work, in general, can be accepted.